# Small G Protein Regulates Virus Infection via MiRNA and Autophagy in Shrimp

**DOI:** 10.3390/biom15020277

**Published:** 2025-02-13

**Authors:** Yaodong He, Yiqi Hu, Ting Ye

**Affiliations:** 1School of Fisheries, Zhejiang Ocean University, Zhoushan 316022, China; yaodong@zjou.edu.cn; 2College of Life Sciences and Medicine, Zhejiang Sci-Tech University, Hangzhou 310018, China; yiqihu@zju.edu.cn

**Keywords:** shrimp, Rab10, miR-2c, autophagy, antiviral

## Abstract

Recently, there has been a burgeoning scholarly interest in elucidating the functional significance and regulatory mechanisms underlying the involvement of small G proteins, such as Rab, in the antiviral immune response of crustaceans. Rab is a member of the small G protein family and plays a crucial role in the transport of cell membranes within eukaryotic cells. It is involved in the movement of cell membranes both within the cell and on its surface, aiding in the entry of effector proteins into specific membrane subregions. While previous research has highlighted the importance of Rab in phagosome formation and maturation, as well as the clearance of innate immune pathogens by phagocytes, its role in regulating autophagy and the antiviral mechanism remains unclear. This study focused on Rab10 and its role in the autophagy pathway within shrimp, as it pertains to defending against viral infections. MiRNA targeting Rab10 was analyzed and verified by bioinformatic methods. It was found that inhibition of miR-2c could enhance the shrimp’s ability to combat viral infections. This discovery suggests a potential new strategy for screening antiviral drugs. In summation, this investigation augments our comprehension of the antiviral mechanism associated with Rab10, illuminating its significance in the antiviral immune response of shrimp.

## 1. Introduction

The Rab family encompasses over 60 members and serves pivotal functions in endocytosis, extracellular membrane trafficking and the recruitment of effector proteins to distinct membrane subdomains [1]. Several members of the Rab family, such as Rab5, Rab6, Rab7 and Rab11, have been implicated in phagocytosis. However, their precise molecular mechanisms in orchestrating phagocytic responses against viral infections remain enigmatic [2,3,4]. Recent investigations into Rab proteins in shrimp have unveiled their pivotal roles in the antiviral responses of shrimp by initiating phagocytic signaling pathways [5]. Specifically, shrimp Rab6 exhibits GTP-binding activity and comprises six GTP-binding domains along with five characteristic Rab domains. Studies have shown elevated expression of Rab6 in WSSV (white spot syndrome virus)-resistant shrimp, and targeted silencing of Rab6 via specific siRNA significantly enhances WSSV replication in shrimp [5]. Rab6 forms a protein complex, leading to actin filament rearrangement and impacting the phagocytosis process, thus playing a pivotal role in shrimp antiviral responses [4,5,6]. In Drosophila, Rab6 regulates actin tissue throughout the phagocytosis process, akin to the early marker Rab5 and the late marker lamp1. The phagocytic function of Rab6 protein in Drosophila melanogaster mirrors its role in shrimp, underscoring the conservation of Rab6’s phagocytic function across invertebrates [7,8]. Additionally, Rab7 serves as a central regulator of lysosomal transport mechanisms [9,10]. Studies have found that shrimp Rab7 directly interacts with the VP28 protein of WSSV, suggesting its involvement in WSSV infection [11]. Silencing the expression of pmrab7 by sequence-specific dsRNA can significantly reduce the expression of wssv-vp28 in shrimp, thus inhibiting the replication of WSSV [12]. Hence, it is plausible that pmrab7 potentially participates in the endosomal transport pathway co-opted by the WSSV to facilitate successful replication. Mounting evidence suggests that members of the Rab family of small G proteins also exert significant regulatory influence over autophagy processes. Both Rab5 and Rab7 are engaged in the upstream regulatory processes and formation of autophagosomes [13]. Rab5 is indispensable for early endocytosis and directly governs the formation of autophagosomes as part of the becn1-pik3c3 complex [14,15,16,17,18], while Rab7, depending on its nuclear endosome function, plays a further role in the maturation and transport of autophagosomes [19,20,21,22]. Rab1 and Rab11 emerge as pivotal molecules in the initial phases of autophagosome formation, facilitating the expansion of phagocytes by sourcing membranes from diverse origins. Moreover, Rab1 plays a distinct role in recruiting several Atg proteins and modulating cell membrane transport to the pre-autophagosomal structure (PAS) [23]. Rab11 orchestrates the maturation trajectory of autophagy and the endosomal pathway by regulating the subcellular distribution of the hook protein [24,25]. Additionally, two Golgi-localized Rabs, Rab9 and Rab33, contribute to autophagy processes. Rab9 notably participates in autophagosome formation [26] and potentially influences autophagy directly by furnishing membrane sources derived from the trans-Golgi network (TGN) to facilitate phagocyte expansion. Rab33b, on the other hand, is implicated in both the early and late stages of autophagy, governing autophagosome formation through its effector molecule Atg16 [27].

Li et al. reported on the function of a novel small G protein, Rab (Rab10), which plays a role in recruiting the necessary adapter for autophagosome engulfment of lipid droplets (LDs) [28]. In hepatocytes induced to undergo autophagy, there was a significant enhancement in Rab10 activity, leading to the augmented recruitment of new autophagic membranes to the surface of lipid droplets (LDs). Disruption of Rab10 function, either through small interfering RNA-mediated knockout or expression of GTPase-deficient mutants, resulted in LD accumulation. LC3 is a specific marker of autophagy, gradually transforming from its cytosolic form (LC3-I) to its membrane-bound lipidated form (LC3-II). This process is used to measure the occurrence of autophagy. Lipidation accelerates the migration of LC3-II bands in Western blotting, allowing for the distinction between membrane-bound and non-membrane-bound forms of LC3 [29,30,31]. Throughout autophagy, the activation of Rab10 is pivotal for LC3 to recruit autophagosomes and facilitate its binding to ehbp1 and ehd2. This interaction critically facilitates the activation of LDs during the process of liver cell lipid phagocytosis.

MicroRNAs (miRNAs) are small non-coding RNAs that exert critical regulatory functions by binding to specific messenger RNAs, thereby modulating gene expression. This regulatory mechanism holds significant implications for various biological processes [32]. Since their initial discovery in Caenorhabditis elegans, a plethora of miRNAs have been identified across a wide spectrum of organisms through computational or experimental methodologies [33]. In mammals, cellular miRNAs have been documented to participate in antiviral defense mechanisms via the interferon network [34]. The roles of miRNAs in virus–host interactions have garnered escalating attention. Studies have elucidated alterations in both host and viral miRNA expression levels during viral infections [[35],[36],[37],], underscoring the profound impact of miRNAs on virus–host dynamics [38]. In the shrimp *Marsupenaeus japonicus*, thirty-five miRNAs were initially characterized, some of which serve as pivotal regulators in the innate immune system. These miRNAs are implicated in diverse processes, including phagocytosis, apoptosis, and phenol oxidase (PO) activation [39,40]. Furthermore, WSSV miRNAs have been identified to target viral genes, thereby augmenting viral infectivity.

The previous research focuses on the mechanism of miRNAs in regulating viral infection in shrimp, particularly in relation to apoptosis and NF-κB pathways. It has been observed that miRNAs derived from the white spot syndrome virus (WSSV) can target host genes (e.g., WSSV-miR-N24-STAT) to evade host antiviral responses. Moreover, WSSV miRNAs can also target viral genes to regulate viral infection [36]. Conversely, host miRNAs can target WSSV genes (e.g., miR-7-wsv477) to inhibit viral pathogenesis [41]. Furthermore, it has been observed that members of the miRNA and Rab families play crucial roles in the pathogenesis of diseases [42,43,44]. Nonetheless, research investigating the regulation of small G proteins by miRNAs and their impact on viral infections remains limited. In this study, the researchers uncovered that miR-2c regulates Rab10, consequently influencing WSSV infection. This investigation aims to explore the connection between small G proteins and miRNAs based on the function of small G proteins.

## 2. Materials and Methods

### 2.1. Experimental Organisms and LPS Challenge

Healthy specimens of *Marsupenaeus japonicus* (average weight 15 ± 2 g, carapace length 10–12 cm) were procured from the Jinjiang Aquatic Products Market (Gongshu District, Hangzhou, China). Prior to experimentation, all shrimp underwent PCR screening for white spot syndrome virus (WSSV) using appendage-derived genomic DNA Primers (WSSV-F→5′-TATTGTCTCTCCTGACGTAC-3′, WSSV-R→5′-CACATTCTTCACGAGTCTAC-3′). Twenty WSSV-negative individuals were randomly assigned to each experimental group.

The WSSV inoculum (GenBank accession: AF332093.3) was prepared from laboratory-maintained viral stocks [45] propagated in High Five™ cells (Thermo Fisher Scientific, Waltham, MA, USA). For the immune challenge, shrimp received 100 μL intramuscular injections of lipopolysaccharide (LPS) into their third abdominal segment. Gill tissues were collected at predetermined intervals (1, 2, 4, 6 h post-injection) and immediately snap-frozen in liquid nitrogen prior to storage at −80 °C. Rab10 expression dynamics were quantified throughout.

### 2.2. Rapamycin Treatment

To investigate mTOR pathway involvement in Rab10 regulation, shrimp received 200 μL intra-sinus injections of rapamycin (LC Laboratories, Woburn, MA, USA, 50 nM in DMSO/PBS vehicle). Hemolymph samples (200 μL/shrimp) were collected from the pericardial cavity using heparinized syringes at 0, 24, 36 and 48 h post-treatment. Cellular fractions were isolated by centrifugation (800× *g*, 10 min, 4 °C) and lysed in a RIPA buffer containing protease inhibitors (Roche, Basel, Switzerland). Rab10 transcript and protein levels in hemocytes were assessed through the following methods.

### 2.3. WSSV Detection

The PCR was carried out using Primers (WSSV-F→5′-TATTGTCTCTCCTGACGTAC-3′, WSSV-R→5′-CACATTCTTCACGAGTCTAC-3′) with 2×Taq Plus MasterMix (Dye) (CWBIO, Taizhou, China). The annealing temperature was 52 °C.

### 2.4. MiRNA Target Prediction and Validation

The seed sequence, encompassing nucleotides 2-8 at the 5′ terminus of mature miRNAs, mediates target recognition through complementary binding to 3′-untranslated regions (3′-UTRs) of mRNAs. Complete seed complementarity (≥7mer-m8) typically induces mRNA deadenylation and degradation via the CCR4-NOT complex, while partial pairing (6mer) predominantly suppresses translation initiation [46]. To systematically identify Rab10-targeting miRNAs in *Marsupenaeus japonicus*, we employed the following multi-algorithm approach:

RNAhybrid (v2.1.2) calculates minimum free energy (ΔG ≤ −20 kcal/mol) for miRNA:mRNA duplex formation. TargetScan (v7.0) predicts evolutionarily conserved targets using context++ score thresholds (score ≤ −0.3). PITA (v3.0) incorporates target site accessibility with a ΔΔG < −10 kcal/mol cutoff. miRanda (v3.3a) applies the miRanda score ≥140 and energy score ≤−20 kcal/mol.

Prediction parameters were optimized for crustacean genomes by adjusting GC content thresholds (40–60%) and removing vertebrate-centric conservation filters. The Rab10 3′-UTR was scanned using all four algorithms, with candidate miRNAs requiring positive hits in ≥3 prediction tools. Final validation was performed through dual-luciferase reporter assays in S2 insect cells.

### 2.5. Fluorescence Co-Localization

A shrimp hemolymph was laid on a glass slide and fixed with 4% paraformaldehyde for 20 min. After slight shaking and washing with PBS 3 times, the hemolymph was treated with PBST for 10 min to improve membrane permeability. Five percent BSA was used to block the slide for 30 min. A diluted primary antibody was added and then incubated overnight at 4 °C. Then, a slide was washed with PBST three times and fluorescent secondary antibody was added to the slide and incubated for 1 h. The slide was stained with Lyso Tracker Red for 30 min and DAPI staining for 10 min. Lastly, the slide was sealed and detected by Laser confocal microscopy.

### 2.6. RT-PCR Detection of Rab10 Gene Expression

A Vazyme’s HiScript II Q RT SuperMix for qPCR (+g DNA wiper) was used. We followed the instructions in the product manual.

The reaction system for fluorescent quantitative PCR comprised 2 × ChamQ SYBR, qPCR Master Mix 5 μL, forward primer 0.2 μL, reverse primer 0.2 μL, 50 × ROX Reference Dye 1 0.2 μL, cDNA template 1 μL and H_2_O 3.4 μL.

The reaction procedure was as follows: pre-denaturation at 95 °C for 1 min, denaturation at 95 °C for 5 s, annealing and extension at 60 °C for 30 s.

### 2.7. Statistical Analysis

All experimental data were analyzed using SPSS Statistics 22.0 (IBM Corp., Armonk, NY, USA) with rigorous validation through GraphPad Prism 9.0 (San Diego, CA, USA). Following the confirmation of normal distribution via a Shapiro–Wilk test (α = 0.05) and the homogeneity of variances using a Levene’s test, the following parametric analyses were performed:

For pairwise comparisons, a two-tailed Student’s *t*-test with Welch’s correction for unequal variances was utilized.

For multi-group comparisons, we utilized a one-way ANOVA followed by a Tukey’s post hoc test for multiple comparison adjustments.

Non-parametric alternatives (Mann–Whitney U test for two groups; Kruskal–Wallis with Dunn’s correction for multiple groups) were employed when data violated parametric assumptions. Results from ≥3 independent biological replicates (*n* ≥ 5 specimens per replicate) are presented as mean ± SEM. Statistical significance thresholds were established as * *p* < 0.05, ** *p* < 0.01 and *** *p* < 0.001. Raw datasets underwent a Box–Cox transformation to stabilize variance prior to analysis, and effect sizes were calculated using Cohen’s d (for *t*-tests) or η^2^ (for ANOVA).

## 3. Results

### 3.1. The Expression of Rab10 During WSSV Infection

To reveal the impact of viral infection on Rab10 expression in shrimp, muscle tissues were gathered at various times post-infection. Results demonstrated a progressive elevation in Rab10 mRNA expression in shrimp as the infection progressed (Figure 1A). Likewise, a protein expression analysis revealed an increase in Rab10 levels during infection (Figure 1B). Furthermore, the ratio of LC3 I/II indicated enhanced autophagy induced by a WSSV infection. These findings imply a potential role for Rab10 in WSSV infection, potentially facilitating virus replication and influencing virus resistance in shrimp.

To assess the effect of LPS stimulation on Rab10 expression, the gill tissues of shrimp were collected at different time intervals following LPS injection. Rab10 mRNA expression at these time points was evaluated using RT-qPCR (Table 1). An RT-qPCR analysis demonstrated an increase in Rab10 mRNA expression in the shrimp (Figure 1C), reaching a level 2.3 times higher than the control group at 48 h. Furthermore, the protein expression of Rab10 showed a gradual increase with time following the injection (Figure 1D). These findings indicate that Rab10 participated in the innate immune process of shrimp induced by LPS. Endosomes are vesicles that are responsible for internalizing molecules and transferring them from the cell membrane to other cellular compartments. Typically, endocytic complexes (such as the receptor ligand) are assigned to early endosomes, and each component can be transported to new sites (such as lysosome, Golgi apparatus, etc.) through late endosomes. Before interacting with Rab5 GTPase and phosphatidylinositol 3-phosphate, Early Endosomal Antigen 1 (EEA1) serves to anchor endocytic vesicles. The co-localization of Rab10 with EEA1 in the cytoplasm suggests that Rab10 may be involved in vesicular trafficking (Figure 1E).

### 3.2. Role of Rab10 in the Defense Mechanism Against Viral Infections

In this study, an in vitro synthesized Rab10 siRNA was used to knock down Rab10 expression in shrimp. To further elucidate the role of Rab10 in viral infection, shrimp groups were co-injected with WSSV along with Rab10 siRNA scrambled, while another group received WSSV alone as control. Gill tissues from the shrimp were gathered at various times post-infection. As anticipated, the Rab10 mRNA expression levels exhibited a decline over time in the siRNA-treated group (Figure 2A,C). A Western blot analysis further validated a progressive reduction in the Rab10 protein expression within the siRNA-treated group (Figure 2B,D), affirming the effective inhibition of Rab10 gene expression by the siRNA. The increase in viral copies and shrimp mortality was observed at different time points after infection. Compared to other groups, the Rab10 siRNA group showed a higher increase in WSSV copies, indicating the crucial role of Rab10 in antiviral immunity (Figure 2E). The mortality rate of shrimp in the siRNA group was markedly higher than that of the other experimental groups beginning from the third day of infection (Figure 2F). These findings underscore the crucial role of Rab10 in the defense mechanism against viral infections in shrimp.

### 3.3. Regulation of Rab10 by miR-2c

Four prediction methods (MiRanda, Target Scan, Pictar and RNAhybrid) were used to predict the miRNA targeting Rab10. The comprehensive prediction results indicated that Rab10 may be regulated by miR-2c (Figure 3A,B).

MiR-2c was overexpressed by injecting miR-2c-mimic into shrimp, and miR-2c-mimic-scrambled was used as control. The muscle tissues of shrimp were collected 48 h after infection, and the RNA was extracted for reverse transcription. The findings demonstrated that overexpression of miR-2c significantly suppressed Rab10 mRNA expression in shrimp (Figure 3C–E). Moreover, a Western blot analysis revealed a notable inhibition in Rab10 protein expression in shrimp following the protein cleavage of muscle tissue (Figure 3F).

### 3.4. Impact of miR-2c Expression on WSSV Infection

To further investigate the role of miR-2c in viral pathogenesis, miR-2c was overexpressed in shrimp using miR-2c mimics, with scrambled miR-2c mimics serving as the negative control. A quantitative analysis demonstrated a significant upregulation of miR-2c levels in the experimental group (Figure 4A,B). Notably, shrimp overexpressing miR-2c exhibited a substantial increase in WSSV viral load compared to the control group (Figure 4C), which correlated with significantly higher mortality rates (Figure 4D). These observations suggest that miR-2c may facilitate WSSV replication and pathogenesis in shrimp. Conversely, miR-2c expression was suppressed through co-injection of antisense oligodeoxynucleotide targeting miR-2c (amo-miR-2c) with WSSV inoculum (Figure 4E). The control groups received either WSSV alone or WSSV with scrambled amo-miR-2c. A quantitative analysis revealed a significant reduction in viral load in shrimp treated with amo-miR-2c compared to both control groups (Figure 4F). Furthermore, inhibition of miR-2c expression resulted in a marked decrease in virus-induced cumulative mortality (Figure 4G). These results demonstrate that the suppression of miR-2c expression effectively attenuates WSSV infection and improves survival outcomes in shrimp.

Collectively, these findings provide compelling evidence that miR-2c plays a crucial role in modulating WSSV infection, with its overexpression exacerbating viral pathogenesis and its inhibition conferring protective effects against WSSV infection in shrimp.

## 4. Discussion

Our study elucidates two pivotal findings regarding antiviral immunity in crustaceans: (1) Rab10 demonstrates essential antiviral functions against WSSV infection, as evidenced by its upregulated expression post-infection and the exacerbated viral replication/mortality upon its suppression; (2) miR-2c facilitates viral pathogenesis by targeting Rab10, representing the first reported mechanism of crustacean miRNA-mediated regulation through small G protein modulation.

These observations align with emerging evidence that host-encoded miRNAs critically influence viral dynamics through host gene silencing [47,48,49]. While previous studies established miRNA-mediated antiviral regulation in vertebrates [46], our findings extend this paradigm to invertebrate systems by demonstrating miR-2c’s capacity to enhance WSSV pathogenicity through Rab10 suppression. Notably, this contrasts with Rosani et al.’s (2020) report of miRNA-mediated antiviral effects in *Crassostrea gigas* [50], suggesting taxonomic specificity in miRNA regulatory networks.

The mechanistic significance of Rab10 in vesicle trafficking provides a plausible explanation for its antiviral role. As endocytic pathways are frequently exploited by DNA viruses for entry and replication [51], Rab10 upregulation may represent an evolutionary adaptation to restrict viral trafficking. Conversely, miR-2c-mediated Rab10 suppression could facilitate viral propagation by compromising vesicular compartmentalization—a hypothesis supported by our observation of increased virion loads following miR-2c overexpression.

From a translational perspective, the miR-2c/Rab10 axis presents dual therapeutic targets: Enhancing Rab10 expression or inhibiting miR-2c activity could potentially bolster crustacean defenses against the WSSV. This is particularly relevant given aquaculture’s USD 42 billion annual losses to viral outbreaks (The State of World Fisheries and Aquaculture, FAO, 2022). Therapeutically, locked nucleic acid (LNA) antisense oligonucleotides against miR-2c, similar to those successfully deployed in vertebrate models [52], warrant investigation. While establishing miR-2c’s regulatory role, our study did not characterize the complete miRNA-mRNA interaction network in WSSV pathogenesis. Further transcriptomic analyses should identify additional miRNA targets within the Rab protein family. Moreover, in vivo delivery methods for miRNA inhibitors in aquaculture settings require optimization. Future work should also examine whether this regulatory mechanism extends to other crustacean pathogens like the yellow head virus or Taura syndrome virus.

## 5. Conclusions

To sum up, this study clarified that Rab10 regulates host resistance to virus infection by participating in autophagy in shrimp, expanding the field of vision of small G protein in invertebrate immunity research and enriching the knowledge on aquatic invertebrate immunity.

## Figures and Tables

**Figure 1 biomolecules-15-00277-f001:**
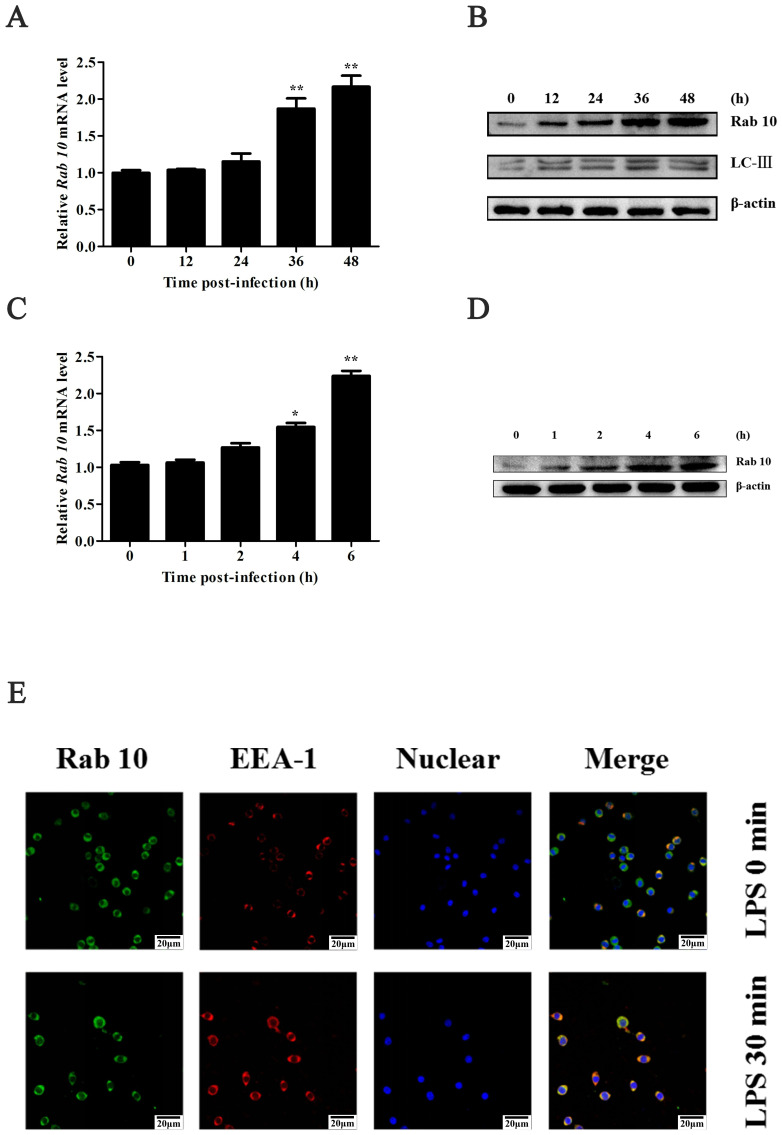
Effect of WSSV infection on the expression of Rab10 (**A**) Expression of Rab10 mRNAs at different times post WSSV infection. (**B**) Western blots of Rab10 and LCIII. (**C**) Expression of Rab10 mRNA during WSSV infection (0–6 h). (**D**) Western results of Rab10 during WSSV infection (0–6 h). (**E**) Confocal of Rab10 and EEA-1 during LPS injection. Staining dye or antibody: Rab10 (Rab antibody), EEA (Lyso-Tracker Red), Nuclear (DAPI). β-actin was used as control. Original western blots can be found at Appendix A. * *p* < 0.05, ** *p* < 0.01.

**Figure 2 biomolecules-15-00277-f002:**
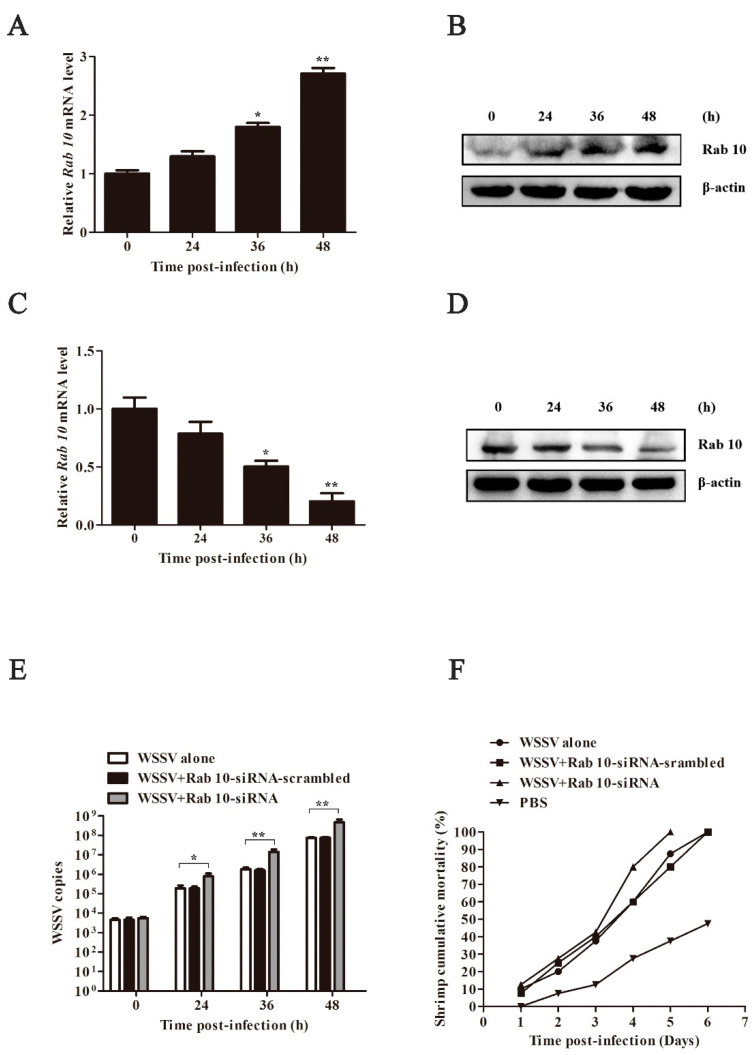
Inhibition of Rab10 in viral infection. Expression of Rab10 mRNAs at different times post WSSV infection with control siRNA (**A**) and Rab10 siRNA (**C**). Western blots of Rab10 at different times post WSSV infection with control siRNA (**B**) and Rab10 siRNA (**D**); β-actin was used as control. Copies of the WSSV (**E**) and mortality of shrimp (**F**) at different times post WSSV infection. Original western blots can be found at Appendix A. * *p* < 0.05, ** *p* < 0.01.

**Figure 3 biomolecules-15-00277-f003:**
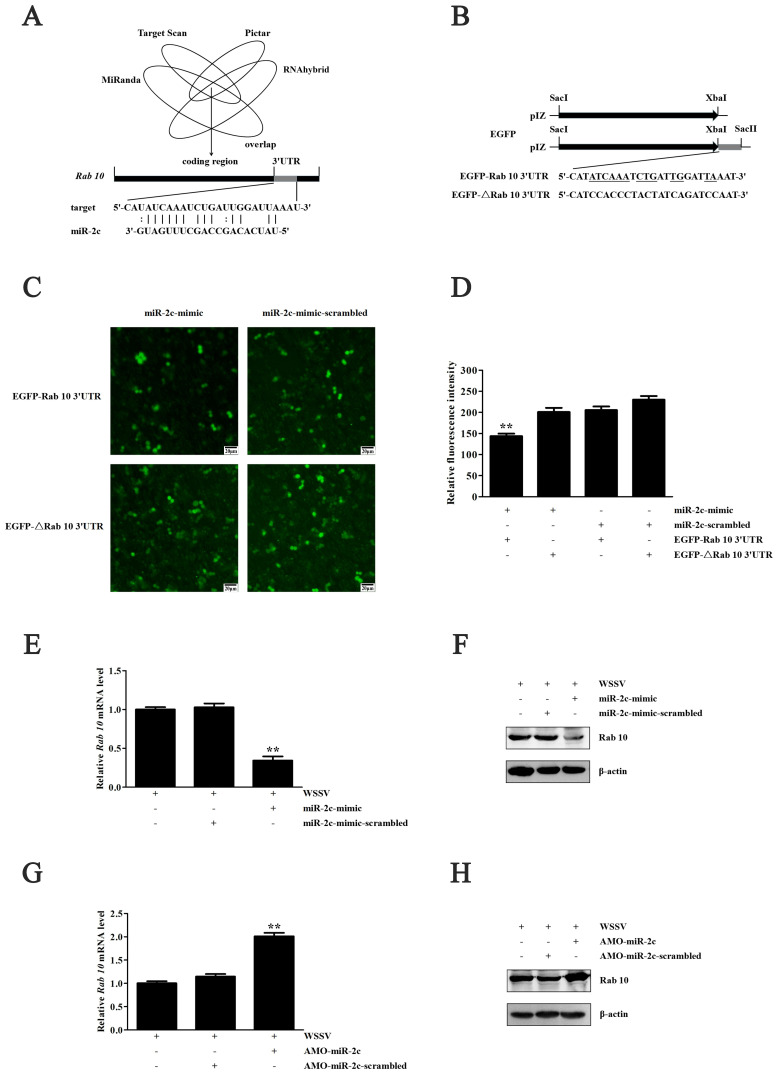
Prediction and validation of the miRNA targeting Rab10. (**A**) Prediction of the miRNA targeting Rab10. (**B**) EGFP plasmid construction. (**C**) The changes in the cellular fluorescence by the miR-2c and Rab10 genes. (**D**) The relative fluorescence (GFP) intensity of High Five cells. The miR-2c expression was overexpressed (**E**) or silenced (**G**) in shrimp. Then, the Western results of Rab10 were examined (**F**,**H**). Original western blots can be found at Appendix A. ** *p* < 0.01.

**Figure 4 biomolecules-15-00277-f004:**
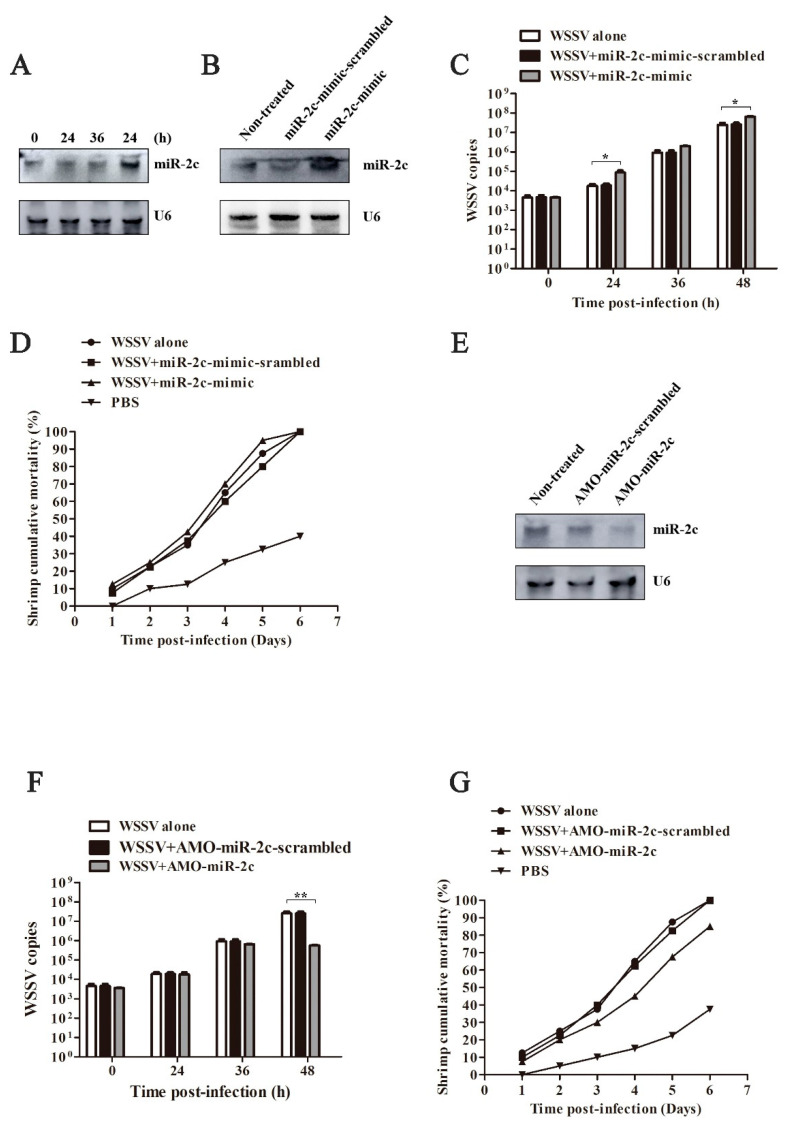
The effect of miR-2c expression on WSSV infection. (**A**) The miR-2c expression in shrimp during viral infection. U6 was used as a control. (**B**) The miR-2c over expression by miR-2c-mimic. (**C**) The effects of miR-2c overexpression on WSSV copies. (**D**) Effects of miR-2c overexpression on shrimp mortality. After different treatments, the mortality was monitored. (**E**) The silencing of miR-2c expression in vivo. (**F**) The effects of miR-2c silencing on WSSV copies. (**G**) The effects of miR-2c silencing on shrimp mortality. After treatments, the mortality was monitored. Original Northern blots can be found at Appendix A. * *p* < 0.05, ** *p* < 0.01.

**Table 1 biomolecules-15-00277-t001:** Primers of qPCR.

Primer	Sequence
Rab10-F	5′-GTAACTTGATCTTCTTTCCTC-3′
Rab10-R	5′-GTTATTCAAACTCCTCCTTAT-3′
GAPDH-F	5′-GGTGCCGAGTACATCGTTGAGTC-3′
GAPDH-R	5′-GGCAGTTGGTAGTGCAAGAGGC-3′

## Data Availability

The authors confirm that the data supporting the findings of this study are available within the article.

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
