# Peer review of "Small G Protein Regulates Virus Infection via MiRNA and Autophagy in Shrimp"

_biomolecules, 2025, doi:10.3390/biom15020277_

Round 1
Reviewer 1 Report
Comments and Suggestions for Authors
Please see the attached file. Thank you

The write should be avoided with spurious statements and unclear conclusions. Thank you
Author Response
We sincerely appreciate the time and effort dedicated by the reviewers to critically evaluate our work. All comments and suggestions have been carefully considered and addressed in detail. All modifications have been highlighted in red font for easy identification. The constructive feedback has significantly strengthened the rigor and clarity of this study. A point-by-point response to each query is provided below, with corresponding revisions highlighted in the updated manuscript.
Detailed Comments
- Para 2.1, Line 3-4: The authors have not mentioned the detailed method used to check that the purchased shrimp are disease-free. They must provide a reference for the PCR method used for this purpose.
Response:Appendages from each shrimp were cut, the DNA was extracted. The PCR was carried out using Primers (WSSV-F 5’-TATTGTCTCTCCTGACGTAC-3’, WSSV-R 5’-CACATTCTTCACGAGTCTAC-3’) with 2×Taq Plus MasterMix (Dye) (CWBIO, China). The annealing temperature is 52°C.
Reference https://doi.org/10.4161/rna.20741
- Para 2.1, Line 2-3: It would be better to provide the mean values of the weight and length of the shrimp used.
Response:Marsupenaeus japonicus (average weight 15 ± 2 g, carapace length 10-12 cm) were used in this research.
- Para 2.2: How was the hemolymph collected? What approach was used? Did they store the whole hemolymph with a diluent (which one?) or were the cells separated by centrifugation? A detailed version of the methodology is required for the manuscript.
Response:Locate the position of the pericardial cavity, which is slightly posterior to the center of the dorsal side of the cephalothorax of the shrimp. Generally, by observing the appearance of the shrimp, a relatively soft and slightly lighter - colored area on the cephalothorax is approximately the position of the pericardial cavity.
Using a sterile syringe filled with an anticoagulant(EDTA), insert the needle vertically or at an angle (about 45°) into the pericardial cavity from above. Then, slowly aspirate the hemolymph. Note that the movements should be gentle to avoid damaging the internal organs of the prawn. The aspiration volume depends on the experimental requirements. Generally, 0.1 - 0.5 mL of hemolymph can be collected from each shrimp.
- Para 2.2: Why did the authors mention ‘shrimp blood lymphocytes’? Hemolymph cells (hemocytes) have already been characterized in shrimp, so they should use a more technical term for the cells used.
Response:‘Shrimp blood lymphocytes’ is now changed to shrimp hemocytes.
- Para 2.3: What is the reference for these primers? The authors are requested to rewrite the methodology in a more detailed and scientific manner (not exclusively for Para 2.3, but for the entire methodology section). The current writing lacks sufficient data for the repeatability of the experiments.
Response:The MM part was rewritten. Para 2.1-2.7
- Statistics: What statistical analysis was used for the data in this manuscript? Please mention this in the relevant part of the manuscript.
Response:All experimental data were analyzed using SPSS Statistics 22.0 (IBM Corp., Armonk, NY, USA) with rigorous validation through GraphPad Prism 9.0 (San Diego, CA, USA). Following confirmation of normal distribution via Shapiro-Wilk test (α = 0.05) and homogeneity of variances using Levene's test, parametric analyses were performed: Pairwise comparisons: Two-tailed Student's t-test with Welch's correction for unequal variances; Multi-group comparisons: One-way ANOVA followed by Tukey's post hoc test for multiple comparisons adjustment; Non-parametric alternatives (Mann-Whitney U test for two groups; Kruskal-Wallis with Dunn's correction for multiple groups) were employed when data violated parametric assumptions. Results from ≥3 independent biological replicates (n ≥ 5 specimens per replicate) are presented as mean ± SEM. Statistical significance thresholds were established as *p < 0.05, **p < 0.01, and ***p < 0.001. Raw datasets underwent Box-Cox transformation to stabilize variance prior to analysis, and effect sizes were calculated using Cohen's d (for t-tests) or η² (for ANOVA).
- Para 3.1: Please indicate which result figure supports the statement, "Furthermore, the ratio of LC3I/II... WSSV infection." Is Fig. 1B showing two bands in the western blot? Please comment on this and cite the figure.
Response:The autophagy protein LC3 (Microtubule - associated protein 1A/1B - light chain 3) is a protein that plays a crucial role in the process of autophagy. Autophagy is a highly conserved catabolic process within cells. It can eliminate damaged organelles, misfolded proteins, and invading pathogens in cells, which is essential for maintaining the stability of the intracellular environment and the normal functions of cells.
Initially, LC3 exists in an inactive precursor form (pro - LC3). After translation, it is cleaved by the cysteine protease ATG4, removing several amino acids at the C - terminus to form the cytoplasmic soluble form, LC3 - I.
LC3 - I can be further modified through a ubiquitin - like conjugation system. With the action of ATG7and ATG3, LC3 - I covalently binds to phosphatidylethanolamine (PE) and is converted into the membrane - bound form, LC3 - II. LC3 - II can specifically localize on the autophagosome membrane, so it is often used as a marker for autophagosome formation and autophagic activity.
- Fig 1E and 3C: The images lack proper scale bars for the cells. Please add these to the figures. Also, include the dye used for staining in the image titles for better understanding.
Response:scale bars and the dye used for staining were added to the Fig 1E and 3C. Figure 1&3 and figure Captions.
- Para 3.3: What input values were used to predict the miRNA targeting Rab10? The authors are requested to provide a detailed analysis of the results shown in Fig. 3A and 3B.
Response:The seed sequence, encompassing nucleotides 2-8 at the 5' terminus of mature miRNAs, mediates target recognition through complementary binding to 3'-untranslated regions (3'-UTRs) of mRNAs. Complete seed complementarity (≥7mer-m8) typically induces mRNA deadenylation and degradation via the CCR4-NOT complex, while partial pairing (6mer) predominantly suppresses translation initiation (Bartel, 2018). To systematically identify Rab10-targeting miRNAs in Marsupenaeus japonicus, we employed a multi-algorithm approach:
RNAhybrid (v2.1.2): Calculates minimum free energy (ΔG ≤ -20 kcal/mol) for miRNA: mRNA duplex formation. TargetScan (v7.0): Predicts evolutionarily conserved targets using context++ score thresholds (score ≤ -0.3).PITA (v3.0): Incorporates target site accessibility with ΔΔG < -10 kcal/mol cutoff. miRanda (v3.3a): Applies miRanda score ≥140 and energy score ≤ -20 kcal/mol.
Prediction parameters were optimized for crustacean genomes by adjusting GC content thresholds (40-60%) and removing vertebrate-centric conservation filters. The Rab10 3'-UTRwas scanned using all four algorithms, with candidate miRNAs requiring positive hits in ≥3 prediction tools. Final validation was performed through dual-luciferase reporter assays in S2 insect cells.
- Para 3.4: The authors state that they overexpressed miR-2c in shrimp compared to the control. The results indicate a significant increase in miR-2c content in shrimp. It is expected that overexpression of miR-2c would increase its content. The authors need to clarify what they are trying to convey, as the current writing is unclear. It is recommended to rewrite these results for better clarity.
Response:Result of Fig.4 was rewritten.
- Statistical Analysis: The authors must perform statistical analysis of the data presented and indicate how significantly different the results are from the control data. This should be reflected in the figures and mentioned in the text as a finding. Otherwise, the results may not be convincing to readers.
Response:All experimental data were analyzed using SPSS Statistics 22.0 (IBM Corp., Armonk, NY, USA) with rigorous validation through GraphPad Prism 9.0 (San Diego, CA, USA). Following confirmation of normal distribution via Shapiro-Wilk test (α = 0.05) and homogeneity of variances using Levene's test, parametric analyses were performed:
Pairwise comparisons: Two-tailed Student's t-test with Welch's correction for unequal variances
Multi-group comparisons: One-way ANOVA followed by Tukey's post hoc test for multiple comparisons adjustment; non-parametric alternatives (Mann-Whitney U test for two groups; Kruskal-Wallis with Dunn's correction for multiple groups) were employed when data violated parametric assumptions. Results from ≥3 independent biological replicates (n ≥ 5 specimens per replicate) are presented as mean ± SEM. Statistical significance thresholds were established as *p < 0.05, **p < 0.01, and ***p < 0.001. Raw datasets underwent Box-Cox transformation to stabilize variance prior to analysis, and effect sizes were calculated using Cohen's d (for t-tests) or η² (for ANOVA).
- Discussion: The discussion must be more explorative based on the manuscript's results. Many of the points discussed are already mentioned in the introduction. The current work appears to be more of a verification study rather than a detailed molecular exploration. The discussion lacks detailed comments and comparisons with existing research on the relevant topic. The authors are recommended to revisit the discussion section in a more comprehensive manner.
Response:We have rewritten the Discussion section as the reviewer`s suggestion.
Reviewer 2 Report
Comments and Suggestions for Authors
In this study, the authors attempt to explore the connection between small G proteins (Rab10) and miRNAs based on the function of small G proteins. The manuscript sheds light on an interesting data. However, it has several shortcomings, and some important points must be clarified or fixed.
I strongly encourage the authors to address the following points:
1. Detection of WSSV by PCR. What are the steps and materials needed to conduct a PCR?
2. Kindly provide the B-actin sequence primer.
3. Which statistical method did the author employ in each experiment? The results should be shown in your picture.
4. Could you kindly provide more information regarding the process of detecting the WSSV copy number? The WSSV gene that was detected.
5. The expression of Rab10 is shown in Fig. 1C in Gill tissue but in Fig. 1A in shrimp muscle that is different. Why?
6. Why did the author of miRNA target prediction use four different ways to find the miRNA that targets Rab10? Please explain your data and what you discussed about with this miRNA in more detail.
Author Response
We sincerely appreciate the time and effort dedicated by the reviewers to critically evaluate our work. All comments and suggestions have been carefully considered and addressed in detail. All modifications have been highlighted in red font for easy identification. The constructive feedback has significantly strengthened the rigor and clarity of this study. A point-by-point response to each query is provided below, with corresponding revisions highlighted in the updated manuscript.
I strongly encourage the authors to address the following points:
- Detection of WSSV by PCR. What are the steps and materials needed to conduct a PCR?
Response:Appendages from each shrimp were cut, the DNA was extracted. The PCR was carried out using Primers (WSSV-F 5’-TATTGTCTCTCCTGACGTAC-3’, WSSV-R 5’-CACATTCTTCACGAGTCTAC-3’) with 2×Taq Plus MasterMix (Dye) (CWBIO, China). The annealing temperature is 52°C.
- Kindly provide the B-actin sequence primer.
Response:B-actin was used as control for western blots, the purchased actin antibody (CAT#: Beyotime A0101) was used as the primary antibody.
- Which statistical method did the author employ in each experiment? The results should be shown in your picture.
Response:All experimental data were analyzed using SPSS Statistics 22.0 (IBM Corp., Armonk, NY, USA) with rigorous validation through GraphPad Prism 9.0 (San Diego, CA, USA). Following confirmation of normal distribution via Shapiro-Wilk test (α = 0.05) and homogeneity of variances using Levene's test, parametric analyses were performed: Pairwise comparisons: Two-tailed Student's t-test with Welch's correction for unequal variances; Multi-group comparisons: One-way ANOVA followed by Tukey's post hoc test for multiple comparisons adjustment; Non-parametric alternatives (Mann-Whitney U test for two groups; Kruskal-Wallis with Dunn's correction for multiple groups) were employed when data violated parametric assumptions. Results from ≥3 independent biological replicates (n ≥ 5 specimens per replicate) are presented as mean ± SEM. Statistical significance thresholds were established as *p < 0.05, **p < 0.01, and ***p < 0.001. Raw datasets underwent Box-Cox transformation to stabilize variance prior to analysis, and effect sizes were calculated using Cohen's d (for t-tests) or η² (for ANOVA).
- Could you kindly provide more information regarding the process of detecting the WSSV copy number? The WSSV gene that was detected.
Response:To quantify the copies of WSSV virions in shrimp, realtime PCR (RT-PCR) was performed using WSSV-specific primers WSSV-specific-F(5’-TTGGTTTCATGCCCGAGATT-3’,WSSV-specific-R 5’-CCTTGGTCAGCCCCTTGA-3’) and a TaqMan fluorogenic probe (5'-FAM-TGC TGC CGT CTC CAA-TAMRA -3') as described previously.32 A plasmid containing a 1400-bp DNA fragment from the WSSV genome was used as the reference plasmid. The PCR reaction mixture (25 μL) consisted of the WSSV genomic DNA aliquot, 200 nM of each primer, 100 nM of each TaqMan probe, and 1 × PCR reaction buffer containing DNA polymerase. PCR amplification was performed for 30 sec at 95°C followed by 45 cycles of 10 sec at 95°C for, 30 sec at 52°C, and 30 sec at 72°C.
Reference https://doi.org/10.4161/rna.20741
- The expression of Rab10 is shown in Fig. 1C in Gill tissue but in Fig. 1A in shrimp muscle that is different. Why?
Response:Due to the rich hemolymph in the gills, the virus spreads rapidly. For the short - term infection (with 6h) results, we used the data from the gills. For the long - term infection effects, we used the data from the muscles.
- Why did the author of miRNA target prediction use four different ways to find the miRNA that targets Rab10? Please explain your data and what you discussed about with this miRNA in more detail.
Response:Each of the miRNA target prediction methods presents distinct advantages and limitations. The selection of an appropriate method is contingent upon the specific research objectives and the availability of relevant data. The prediction method founded on sequence conservation is particularly well - suited for scenarios where the sequences of known miRNA targets exhibit a relatively high degree of conservation across species. The method relying on complementary matching is characterized by its computational simplicity, as it primarily focuses on the base - pairing rules between miRNAs and their potential target sites. The machine - learning - based prediction method holds significant promise in enhancing the accuracy of miRNA target prediction. By leveraging large - scale experimental datasets, machine - learning algorithms can learn complex patterns and relationships that may not be readily discernible through traditional methods. Nevertheless, the implementation of this approach demands substantial computational resources and a comprehensive set of training data.
Therefore, predicting miRNA targets using multiple methods and then selecting the intersection of the prediction results can significantly improve the accuracy and efficiency of the prediction.
We have also updated the miRNA target prediction in M&M section.
Round 2
Reviewer 1 Report
Comments and Suggestions for Authors
Please see the attached file. Thank you

Issues in spacing, writing format
Author Response
Detailed Comments
- The full virus name should be given in line 39 where the authors mentioned WSSV in the manuscript. This allows the authors to delete the full name of WSSV in line 112. Also, for your information, virus names should be written in lowercase letters, with no need to capitalize the first letter of each word unless it is a place name or other specific noun.
Response: Thank you for your question. The full name of WSSV is White Spot Syndrome Virus.
WSSV is a proper noun, and it is conventionally represented in uppercase letters to denote this virus.
- In lines 112-113: What do the authors mean by appendage-derived genomic DNA primers?
Response: The meaning of this sentence is to perform PCR amplification on the collected appendages using viral genome primers.
- Lines 116-117: Please cite the reference for the WSSV inoculum preparation or briefly explain it in the materials and methods section if the manuscript made any deviations.
Response: WSSV inoculum was prepared according to the following reference:
Wang X W , Xu Y H , Xu J D ,et al.Collaboration between a soluble C-type lectin and calreticulin facilitates white spot syndrome virus infection in shrimp. Journal of Immunology, 2014, 193(5):2106-2117.DOI:10.4049/jimmunol.1400552.
- Line 126: Please change ‘800 × g’ to ‘800 × g’. ‘g’ should be italics.
Response: ‘g’ is now changed to italics.
- Lines 126, 13, 151, 152, 125, 119, etc.: Degree Celsius and time (hour) units written in the manuscript have some disparity. Please consider correcting them in a consistent manner.
Response: Thank you for your suggestion. Degree Celsius and time (hour) units have been corrected.
- Please review the subheadings and correct them to follow a single style, i.e., either capitalize each word or not.
Response: Thank you for your suggestion. We have corrected subheadings of Para 2.1-2.7, 3.4.
- Line 155: RT-PCR or RT-qPCR? Please state it correctly.
Response: It`s RT-qPCR. The original text has also been modified.
- Figures 1A, C, 2A, C, E, 3D, E, F; 4C, F lack statistical significance indications. The authors mentioned in the materials and methods section (lines 174-175) that they performed statistical analyses to determine the significance of their results. Kindly check the figures and indicate the significance compared to the control data.
Response: Significance indications were added to Figures 1A, C, 2A, C, E, 3D, E, F; 4C, F (*p < 0.05, **p < 0.01, and ***p < 0.001).
- Overall, the manuscript has some writing issues such as spacing, proper usage of unit placement, and maintaining a consistent format in subheadings. Kindly review the manuscript once again and correct these issues accordingly.
Response: We greatly appreciate the reviewer's comments and suggestions, and we have made revisions and corrections to the corresponding issues.
Reviewer 2 Report
Comments and Suggestions for Authors
-
Author Response
We greatly appreciate the reviewer's comments and suggestions, and we have made revisions and corrections to the corresponding issues.